# Nutrition Intervention for Reduction of Cardiovascular Risk in African Americans Using the 2019 American College of Cardiology/American Heart Association Primary Prevention Guidelines

**DOI:** 10.3390/nu13103422

**Published:** 2021-09-28

**Authors:** Kim Allan Williams, Ibtihaj Fughhi, Setri Fugar, Monica Mazur, Sharon Gates, Stephen Sawyer, Hena Patel, Darrius Chambers, Ronald McDaniel, Jochen R. Reiser, Terry Mason

**Affiliations:** 1Division of Cardiology, Rush University Medical Center, 1717 W. Congress Parkway, Suite 303 Kellogg, Chicago, IL 60612, USA; Ibtihaj_A_Fughhi@rush.edu (I.F.); Setri_S_Fugar@rush.edu (S.F.); Monica_Mazur@rush.edu (M.M.); Sharon_Gates@rush.edu (S.G.); Stephen_W_Sawyer@rush.edu (S.S.); Darrius_Chambers@rush.edu (D.C.); Jochen_Reiser@rush.edu (J.R.R.); 2Section of Cardiology, Department of Medicine, University of Chicago, Chicago, IL 60612, USA; henapatel87@gmail.com; 3Boston Heart Diagnostics, Framingham, MA 60637, USA; ronmc44@gmail.com; 4Cook County Department of Public Health, Chicago, IL 60612, USA; terrymasonmd@gmail.com

**Keywords:** African Americans, cardiovascular risk, nutrition intervention, plant-based diet, vegan diet

## Abstract

Introduction: The 2019 American College of Cardiology/American Heart Association (ACC/AHA) Prevention Guidelines emphasize reduction in dietary sodium, cholesterol, refined carbohydrates, saturated fat and sweetened beverages. We hypothesized that implementing this dietary pattern could reduce cardiovascular risk in a cohort of volunteers in an urban African American (AA) community church, during a 5-week ACC/AHA-styled nutrition intervention, assessed by measuring risk markers and adherence, called HEART-LENS (Helping Everyone Assess Risk Today Lenten Nutrition Study). Methods: The study population consisted of 53 volunteers who committed to eat only home-delivered non-dairy vegetarian meals (average daily calories 1155, sodium 1285 mg, cholesterol 0 mg; 58% carbohydrate, 17% protein, 25% fat). Body mass index (BMI) and fasting serum markers of cardiometabolic and risk factors were measured, with collection of any dietary deviation. Results: Of 53 volunteers, 44 (mean age 60.2 years, 37 women) completed the trial (88%); 1 was intolerant of the meals, 1 completed both blood draws but did not eat delivered food, and 7 did not return for the tests. Adherence to the diet was reported at 93% in the remaining 44. Cardiometabolic risk factors improved significantly, highlighted by a marked reduction in serum insulin (−43%, *p* = 0.000), hemoglobin A1c (6.2% to 6.0%, *p* = 0.000), weight and BMI (−10.2 lbs, 33 to 31 kg/m^2^, *p* = 0.000), but with small reductions of fasting glucose (−6%, *p* = 0.405) and triglyceride levels (−4%, *p* = 0.408). Additionally, improved were trimethylamine-N-oxide (5.1 to 2.9 µmol/L, −43%, *p* = 0.001), small dense low-density lipoprotein cholesterol (LDL) (24.2 to 19.1 mg/dL, −21%, *p* = 0.000), LDL (121 to 104 mg/dL, −14%, *p* = 0.000), total cholesterol (TC) (190 to 168 mg/dL, −12%, *p* = 0.000), and lipoprotein (a) (LP(a)) (56 to 51 mg/dL, −11%, *p* = 0.000); high sensitivity C-reactive protein (hs-CRP) was widely variable but reduced by 16% (2.5 to 2.1 ng/mL, *p* = NS) in 40 subjects without inflammatory conditions. Soluble urokinase plasminogen activator (suPAR) levels were not significantly changed. The ACC/AHA pooled cohort atherosclerotic cardiovascular disease (ASCVD) risk scores were calculated for 41 and 36 volunteers, respectively, as the ASCVD risk could not be calculated for 3 subjects with low lipid fractions at baseline and 8 subjects after intervention (*p* = 0.184). In the remaining subjects, the mean 10-year risk was reduced from 10.8 to 8.7%, a 19.4% decrease (*p* = 0.006), primarily due to a 14% decrease in low-density lipoprotein cholesterol and a 10 mm Hg (6%) reduction in systolic blood pressure. Conclusions: In this prospective 5-week non-dairy vegetarian nutrition intervention with good adherence consistent with the 2019 ACC/AHA Guidelines in an at-risk AA population, markers of cardiovascular risk, cardiometabolism, and body weight were significantly reduced, including obesity, low-density lipoprotein cholesterol (LDLc) density, LP(a), inflammation, and ingestion of substrates mediating production of trimethylamine-N-oxide (TMAO). Albeit reduced, hs-CRP and suPAR, were not lowered consistently. This induced a significant decrease in the 10-year ASCVD risk in this AA cohort. If widely adopted, this could dramatically reduce and possibly eradicate, the racial disparity in ASCVD events and mortality, if 19% of the 21% increase is eliminated by this lifestyle change.

## 1. Introduction

Heart disease has been the leading cause of death in the United States since 1918. It is the leading cause of death for people of most racial/ethnic groups in the United States, including African Americans (AAs), Hispanics, and whites, with an estimated cost of over $200 billion annually, in health care services, medications, and lost productivity. After 4 decades of decline, heart disease deaths rose in 2015 by 1%, and has continued to climb. This trend has been attributed to the obesity epidemic and the consequent rise in type II diabetes [1].

Recent literature has focused on several eating patterns, food substances, micronutrients and metabolic consequences in relationship to the development of cardiovascular events and mortality [2]. Many AAs in the United States have a nutrition pattern characterized by high levels of sodium, saturated fat, cholesterol, and refined carbohydrates including sugar. This pattern underpins the high rates of obesity, hypertension, hyperlipidemia and type II diabetes mellitus, and the consequent myocardial infarction, heart failure, stroke, renal disease and cardiovascular mortality [2,3,4,5,6].

Exclusively plant-based diets (often called “vegan”) could present a disease modifying alternative for this at-risk population, as they have been shown to be nutritionally robust, and are widely considered as an option to reduce the incidence or impact of cardiovascular diseases and risk factors, as promoted by the American Dietetic Association [7]. This type of dietary change has been associated with improved weight and lipids AAs in observational studies [8].

The 2019 American College of Cardiology/American Heart Association (ACC/AHA) Primary Prevention Guidelines were designed specifically to reduce cardiovascular risk by recommending reduction in dietary sodium, cholesterol, refined carbohydrates, saturated fat and sweetened beverages [6].

The purpose of this study was to determine the impact of a low-sodium, low-calorie, zero cholesterol, non-dairy vegetarian diet on serum metabolic markers, and the 10-year ACC/AHA pooled cohort atherosclerotic cardiovascular disease (ASCVD) calculator for ASCVD risk estimation in a Chicago AA population. We hypothesized that a 5-week plant-based nutrition intervention with dietary elements consistent with 2019 ACC/AHA Primary Prevention Guidelines, could substantially lower the ASCVD risk in this population. Further, this study aimed to evaluate the impact of a nutrition intervention in AAs on risk factors and biomarkers associated with CVD.

## 2. Methods

### 2.1. Population

All subjects gave their informed consent for inclusion before they participated in the study. The study was conducted in accordance with the Declaration of Helsinki, and the protocol was approved by the Ethics Committee of Rush University (approval code 19020103-IRB01). Eligibility criteria included age of 18 or over, with the ability to consent to dietary intervention, and willingness to comply with eating only the provided meals, but keeping a record of any other food intake if it occurred. Patients were ineligible if they were not eating a standard American (nonvegetarian diet), or if they were unable to regularly measure blood pressure if taking antihypertensive medications, unable to regularly measure blood sugar if taking antihyperglycemic medications, celiac disease (gluten enteropathy), undergoing treatment for eating disorder, active treatment for malignancy, were unable to eat solid food, or had a fruit, nut or vegetable allergy.

Demographic data (including age, gender, height, weight, race/ethnicity, medications currently being taken, risk factors for heart disease (i.e., diabetes, hypertension, hyperlipidemia, and smoking), family history of heart disease, and current medical conditions were obtained (Table 1). The baseline dietary patterns and lifestyle variables of participants were collected using a self-reported questionnaire. Each participant underwent a follow up phone or email questionnaire 4 to 6 months after the intervention to determine if they used the health information obtained and maintained improvement in their diet. Subjects with missing information on essential variables, such as lipid profile and serum biomarkers, were excluded from the analytic sample for that variable.

A total of 53 volunteers agreed to a 5-week prospective dietary intervention performed during Lent in a predominantly AA church on the south side of Chicago (Trinity United Church of Christ, https://www.trinitychicago.org/ (accessed on 22 September 2021)). The population included 7 men and 46 women, mean age 61.3 ± 10.7 years; 26 were hypertensive and 9 had treated diabetes mellitus.

### 2.2. Nutrition Intervention

The patients received fully prepared frozen preservative and cholesterol free, organic whole food plant-based meals (http://veestro.com (accessed on 22 September 2021)) for 5 weeks of 3 meals daily. These meals required either stove, oven or microwave heating prior to eating. All meals were low in saturated fat (6 gm), calories (1155 kc) and sodium (1285 mg) per day, with zero cholesterol (100% plant-based). The calorie distribution averaged 58% from carbohydrates, 18% from protein and 26% from fat (see Appendix A).

### 2.3. Measurements

Heart rate, blood pressure, height, weight, body mass index (BMI), TC, small dense LDLc (sdLDL), LDL, high density lipoprotein (HDL), non-HDL cholesterol (non-HDL), triglycerides (TG), high-sensitivity C-reactive protein (hs-CRP), soluble urokinase plasminogen activator receptor (suPAR) and trimethylamine-N-oxide (TMAO), were obtained prior to and on the last day of the intervention.

### 2.4. Statistical Analysis

The ACC/AHA ASCVD risk estimator plus was used to calculate baseline and follow-up 10-year and lifetime risks of ASCVD (defined as coronary death or nonfatal myocardial infarction, or fatal or nonfatal stroke) event risk. Baseline and follow-up biomarkers, metabolic parameters, blood pressure, weight, ACC/AHA pooled cohort ASCVD risk scores were compared using Student’s *t*-test for paired samples. Data is presented as mean and standard deviation, with the exception of hs-CRP which was analyzed using the median and interquartile range, after censoring any markedly elevated levels (>10 ng/dL) pre- or post-intervention, likely indicating acute infection. The statistical analyses were be performed using SPSS version 22 software (SPSS, Inc., Chicago, IL, USA).

Power analysis was performed using the hypothesis that there would be a statistically significant reduction in the novel marker TMAO (e.g., a 1.5 micromol), with an alpha of 0.05. The minimum sample size, given a margin of error of at most 1.5 micromol, would be 31 patients, with a power of 0.51, assuming a margin of error of 1.5 and a standard error of 0.75.

## 3. Results

Of the 53 volunteers, 44 (mean age 60.2 years, 37 women), completed the trial (88%). One withdrew stating that she was intolerant (“hives”) of the meals, 1 completed both blood draws and risk assessments, but was excluded after she stated that she did not eat the delivered food, and 7 did not return for the follow-up tests. Adherence to the diet was reported at 93% in the remaining 44. Of these 44 subjects, 25 reported use of antihypertensive medication, 11 were on lipid lowering drugs and 7 were taking hypoglycemic agents.

As in Figure 1, measured cardiometabolic factors improved significantly, highlighted by a marked reduction in serum insulin (−43%, *p* = 0.000), hemoglobin A1c (6.2% to 6.0%, *p* = 0.000), weight and BMI (−10.2 lbs, 33 to 31 kg/m^2^, *p* = 0.000), but with small reductions of fasting glucose (107 to 101 mg/dL, *p* = 0.405) and TG levels (−4%, *p* = 0.408).

There was marked improvement in most risk factors (Figure 1, Table 2), including a reduction in TMAO by 43% (5.1 to 2.9 µmol/L, *p* = 0.001), sdLDL by 21% (24.2 to 19.1 mg/dL, *p* = 0.000), LDL by 14% (121 to 108 mg/dL, *p* = 0.000), TC by 12% (190 to 168 mg/dL, *p* = 0.000), and LP(a) by 10% (56 to 51 mg/dL, *p* = 0.000). Serum hs-CRP was widely variable but reduced by 16% (2.5 to 2.1 ng/mL, *p* = NS) in the 40 subjects without inflammatory conditions. Similarly, decreases in TG and suPAR, each by 4% were not significant. Although HDL was reduced by 11%, non-HDL was reduced by 12% (63 to 56 and 127 to 112 mg/dL, respectively, both *p* = 0.000).

Baseline and follow up ASCVD risk scores were calculated for 41 and 36 volunteers, respectively, as the ASCVD risk could not be calculated for 3 subjects with low point of care lipid fractions at baseline and 8 subjects after intervention (TC < 100 mg/dl or TG < 45). In the remaining cohort, the mean 10-year risk was reduced from 10.8 to 8.7%, a 19.4% decrease (*p* = 0.006). This was based primarily on a 14% decrease in LDLc and a 10 mm Hg (6%) reduction in systolic blood pressure (BP).

Out of the 44 participants, only 33 returned the post-study six-month survey. Thirty-one respondents changed their lifestyle as a result of participating in the trial. Of those who changed their lifestyle, 23 continued to eat predominantly plant-based nutrition, 17 began an exercise regimen, 29 stated that they are more mindful of their diet. The frequency of eating exclusively plant-based meals increased from none at baseline, to six participants always eating plant-based, 15 eating plant-based regularly, and only rarely eating plant-based meals in four. The group was asked to identify barriers to eating exclusively plant-based meals, which included (a) difficulty planning meals and finding recipes, (b) inconvenience and poor selections when eating at a restaurant or cooking for a family, and (c) cost of meal delivery and plant-based substitutes. Despite these barriers, most participants reported positive outcomes from this study, in addition to decreased weight, BP and cholesterol, and subjectively increased energy and outlook.

## 4. Discussion

Nutritional factors continue to drive mortality in the US and around the globe. One recent analysis indicated that 11 million premature deaths and 255 million disability-adjusted life years were attributable to dietary risk factors, including high intake of sodium, low intake of whole grains and low intake of fruits. [9] ASCVD remains the leading cause of death in the US, and the cardiovascular disease mortality rate of AAs was 1.21 times that of whites in 2015 [10]. This contributes to the reported up to 30-year difference in life expectancy between some AA (Englewood) versus predominantly Caucasian (Streeterville) neighborhoods in Chicago. Efforts to reduce this disparity have revolved around recognition and treatment of hypertension, but with less attention to evidence-based nutrition intervention.

This short-term, 5-week prospective nutritional intervention in an at-risk AA community demonstrated that cardiovascular risk can be significantly reduced, measured at a 19.4% reduction by the ACC/AHA pooled cohort risk calculator. This is a key population for intervention, since nearly 50% of AA adults have some form of cardiovascular disease, largely mediated by diet, as demonstrated in the REGARDS trial [2]. In that study, the Southern dietary pattern (typified as “soul-food”) was identified as substantially increasing health risks including a 56% higher risk of heart disease and a 30% higher risk of stroke. This pattern consisted of more fried food, added fats, organ and processed meats and sugar sweetened beverages. It has been noted that consuming an unhealthy plant-based diet with similar elements to the Southern diet including juices/sweetened beverages, refined grains, potatoes/fries and sweets results in increased coronary events exceeding the point estimate for that associated with consumption of animal products [11]. A high Southern diet score has been associated with greater degrees of hypertension among black adults in the US, likely a key factor mediating the racial differences in adverse cardiac outcomes, along with the dietary ratio of sodium to potassium, and education level [12].

Correlation between dietary elements and cardiovascular mortality were also examined in a recent publication from the National Health and Nutrition Examination Surveys (NHANES) [13], indicating that high sodium content (>2000 mg daily), red meat (>14 g/day), sugar sweetened beverages or processed red meat consumption, in any amount where associated with cardiovascular death. The relative risk of processed versus unprocessed red meat associated with development of heart failure, coronary heart disease, hemorrhagic stroke, ischemic stroke and diabetes mellitus has been examined in a meta-analysis, indicating that TMAO, heme iron, nitrosamine, nitrates, nitrate and nitroso compounds, saturated fat, advanced glycation end products, and branched amino acids represent possible mediators [14].

Replacing such items in the AA diet in favor of a vegetarian eating pattern reduces or eliminates intake of these risk mediators and has previously been shown observationally to markedly reduce obesity, hypertension and dyslipidemia [8]. However, there is no published prospective nutrition intervention data in this ethnic group on advanced lipid testing or other biomarkers that correlate with cardiovascular events or inflammation, such as suPAR, hs-CRP, TMAO, fasting insulin levels or ASCVD pooled cohort risk scores.

### 4.1. Lipoprotein a (Lp(a))

Dietary interventions have shown variable results on Lp(a) [15,16], and are generally thought to be ineffective. A low-fat diet nonvegetarian diet intervention produced an increase in LP(a) concentration was increased by 7% (*p* < 0.01) with a low-vegetable diet and 9% with a high-vegetable diet [15]. However, in a 4-week plant-based intervention in AAs, Montgomery et al. [16], did show a substantial reduction in serum Lp(a), apolipoprotein B, LDL particles and small-dense LDL cholesterol, very congruent with the 5-week intervention undertaken in this report.

### 4.2. High-Sensitivity C-Reactive Protein (hs-CRP)

Markers of inflammation such as hs-CRP and interleukin-1 have been correlated with adverse cardiac events and mortality [17,18]. While they can be lowered with statins and anti-inflammatory drugs, such as methotrexate or canakinamab, both LDLc and hs-CRP can be lowered with plant-based dietary intervention, as in the Portfolio or other whole food plant-based diet [19,20]. The current study was consistent with their findings, with a 16% decrease in hs-CRP, once the elevated levels (>10 ng/dL) pre- or post-intervention were censored, indicating the anti-inflammatory nature of the plant-based nutrition. However, given the variability of this assay due to common maladies, such as a gingival infection, this decrease did not reach statistical significance.

### 4.3. Soluble Urokinase Plasminogen Activator Receptor (suPAR)

Serum levels of the suPAR is an independent predictor of the development and progression of chronic kidney disease (CKD) [21,22]. AAs have a high frequency (36%) of apolipoprotein L1 (APOL1) gene variants, G1 and G2 [23], that are known to be associated with an increased risk of CKD, but the decline in kidney function associated with APOL1 risk variants has been shown to be dependent on plasma suPAR levels [21]. Although consumption of animal protein is associated with progression of CKD [24], there was no evidence for improvement of suPAR with plant-based dietary intervention.

### 4.4. Insulin

Observational studies indicate that vegetarians have higher insulin sensitivity and lower fasting insulin levels than omnivores independent of BMI, a finding that is more significant in vegetarians who are completely plant-based (“vegans”) [25]. This is important for long-term cardiovascular risk, since elevated insulin levels accelerate accumulation of central obesity, hypertension and atherogenicity of lipids with consequent atherosclerosis [26]. However, no prospective plant-based nutrition intervention trials to reduce insulin levels have been reported. The findings of the current study, a marked 43% decrease in fasting insulin levels, if sustained, would likely provide a marked lowering of plaque burden, weight and subsequent cardiovascular events.

### 4.5. Trimethylamine-N-oxide (TMAO)

Tang et al. [27,28,29] have reported that ingested animal products, with the attendant intake of choline, phosphatidylcholine, creatine and betaine, are chemically converted to trimethylamine by gastrointestinal bacteria species commonly found in omnivores, but not predominant in the microbiome of vegan/vegetarians. Trimethylamine undergoes hepatic oxidation into TMAO, higher levels of which have been associated with a prothrombotic state by stimulating platelet aggregation, as well as with stroke and myocardial infarction, and both heart failure and overall cardiovascular mortality. While interventions to reduce TMAO with alteration of the microbiome or reduction in dietary substrates have been reported, there have been no prospective interventions in an at-risk population. Our study demonstrated a 43% decrease in TMAO, from 5.1 to 2.9 µmol/L, representing a substantial reduction in TMAO-related-risk with plant-based nutrition.

### 4.6. HDL Cholesterol

Consistent with multiple prior publications, HDL was reduced by reduction in animal product consumption. The meta-analysis by Yokohama et al. [30]. found that a plant-based vegetarian diet is associated with a 22.9 mg/dL reduction in LDL cholesterol and a 3.6 mg/dL reduction in HDL cholesterol, compared to control groups following an omnivorous diet. However, in the cited clinical trials, a plant-based diet lowered LDL cholesterol by 12.2 mg/dL and reduced HDL cholesterol by 3.4 mg/dL, compared to control groups following an omnivorous, low-fat, calorie-restricted, or a conventional diabetes diet. Although in our study HDL was reduced by 11% (from 63 to 56), non-HDL was reduced by 12% (63 to 56 and 127 to 112 mg/dL, respectively, both *p* = 0.000), and consequently ASCVD risk was reduced.

### 4.7. Limitations

This study was a short-term intervention, for 5-weeks only and therefore cannot address long-term reduction in risk or measurement of accumulated events. As there was no control group and this was a free-living population, some of the improvement (or lack of improvement of hs-CRP or suPAR) may have been related to non-dietary factors, such as better medication adherence or increased exercise, that were not acknowledged by the subjects. As 8 subjects were removed from the ASCVD risk calculation due to successfully lowered lipid levels (e.g., total cholesterol <130 invalidates to pooled cohort equation), the reduction in the 10-year risk score is likely underestimated. Follow-up was limited to only 75% who returned the surveys but did demonstrate a lasting impact of the intervention. However, the intervention was instituted over a religious holiday season (Lent), which may have increased the likelihood of adherence and long-term success. Lastly, no attempt was made to over-recruit men, who have the highest cardiovascular mortality; taking volunteers first come first serve resulted in 84% female participants. However, given the usual gender disparity of cardiovascular research underrepresenting women, and the higher frequency of obesity in AA women, this could be considered one of the strengths of this project.

## 5. Conclusions

In this prospective 5-week non-dairy vegetarian intervention consistent, with the nutrition section of the 2019 ACC/AHA Primary Prevention Guidelines, in an at-risk AA population, markers of cardiometabolism and body weight, as well as mediators and markers of CVD risk, including fasting insulin, hemoglobin A1c, fasting glucose, lipid fractions, LDL density, LP(a), and production of TMAO were all improved. This resulted in a significant decrease in the 10-year ASCVD risk in this AA cohort. If widely adopted, this could dramatically reduce and possibly eradicate, the racial disparity in ASCVD events and mortality.

Moreover, the current COVID-19 pandemic has had disproportionate effects on the AA population in the US, due to the prevalence of hypertension, diabetes, obesity and elevated cholesterol [31,32]. This has recently been attributed to the intestinal microbiome [33], which has been shown to be improved by the type of plant-based nutritional intervention utilized in this trial [34]. Thus, reduction in these risk factors and improvement of the microbiome are timely, and should undergo rigorous additional longer-term randomized study.

## Figures and Tables

**Figure 1 nutrients-13-03422-f001:**
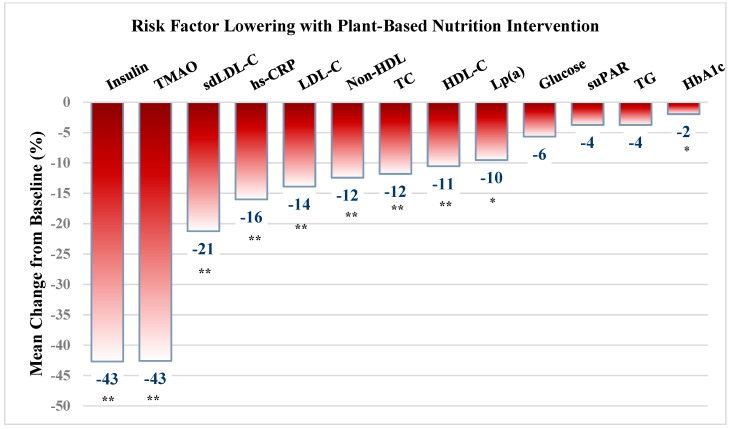
Cardiovascular and cardiometabolic risk factor lowering with plant-based nutrition intervention. Values are presented as mean ± SD. * significant at *p* < 0.01 level; ** significant at <0.001 level. BMI = body mass index; TC = Total cholesterol; sdLDL = small dense LDLc; LDL = Low density lipoprotein, HDL = high density lipoprotein; non-HDL = non-HDL cholesterol; TG = triglycerides; hsCRP = high-sensitivity C-reactive protein; suPAR = soluble urokinase plasminogen activator receptor; TMAO = trimethylamine-N-oxide.

**Table 1 nutrients-13-03422-t001:** Baseline characteristics.

Characteristics	*n* = 44
Age (years) mean ± SD	60 ± 9.6
Gender *n* (%)	
Female	37 (84%)
Male	7 (16%)
Smoking *n* (%)	
Current Smoker	1 (4.7%)
Former Smoker	9 (20.9%)
Never Smoker	33 (76.7%)
Diabetes Mellitus *n* (%)	10 (22.7%), 7 *
Hypertension *n* (%)	25 (57.8%), 25 *
Dyslipidemia *n* (%)	12 (27.3%), 11 *
History of CAD *n* (%)	5 (11.4%)
Family History of CVD *n* (%)	16 (36.4%)

Continuous variables expressed as mean ± standard deviation. Dichotomous variables expressed as frequency (%). * indicates the number treated with medications.

**Table 2 nutrients-13-03422-t002:** Mean Changes in Biomarkers and Risk Factors from Baseline.

Variables	Baseline	Follow Up	*p* Value
TMAO (μM)	5.06 ± 4.21	2.91 ± 1.63	0.001 **
sdLDL-C (mg/dL)	24.2 ± 8.4	19.0 ± 5.9	0.000 **
hs-CRP (mg/L)	2.5 ± 5.2	2.1 ± 2.3	0.128
LDL (mg/dL)	120.8 ± 31.6	104.0 ± 28.4	0.000 **
Non-HDL (mg/dL)	127.3 ± 31.6	111.5 ± 27.4	0.000 **
TC (mg/dL)	190.0 ± 34.0	167.6 ± 30.9	0.000 **
HDL-C (mg/dL)	62.8 ± 16.0	56.1 ± 13.7	0.000 **
Lp(a) (mg/dL)	56.3 ± 44.9	50.9 ± 37.9	0.001 *
suPAR (ng/dL)	3.37 ± 0.84	3.26 ± 1.17	0.66
TG (mg/dL)	79.7 ± 25.5	76.7 ± 21.4	0.408
Insulin (uU/mL)	20.9 ± 38.0	12.0 ± 28.8	0.000 **
Glucose (mg/dL)	107.0 ± 41.9	101.0 ± 39.5	0.405
HbA1c (%)	6.2 ± 1.4	6.0 ± 1.2	0.009 *
Systolic BP (mmHg)	144.6 ± 21.5	135.8 ± 21.3	0.001 *
Diastolic BP (mmHg)	88.4 ± 14.8	85.2 ± 11.5	0.118
Weight (lbs)	198.6 ± 38.6	188.6 ± 36.2	0.000 **
BMI (kg/m^2^)	32.7 ± 6.37971	31.2 ± 5.83631	0.000 **

BMI = body mass index; TC = Total cholesterol; sdLDL = small dense LDLc; LDL = Low density lipoprotein HDL = high density lipoprotein; non-HDL = non-HDL cholesterol; TG = triglycerides; hsCRP = high-sensitivity C-reactive protein; suPAR = soluble urokinase plasminogen activator receptor; TMAO = trimethylamine-N-oxide. Values are presented as mean ± SD. * significant at *p* < 0.01 level; ** significant at <0.001 level.

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
