# Peer review of "Nutrition Intervention for Reduction of Cardiovascular Risk in African Americans Using the 2019 American College of Cardiology/American Heart Association Primary Prevention Guidelines"

_nutrients, 2021, doi:10.3390/nu13103422_

Round 1
Reviewer 1 Report
Dear Authors,
the manuscript presented is well redacted and shown interesting data about a nutrition intervention in a cohort of volunteers in an urban African American community church using the 2019 American College of Cardiology/American Heart Association Primary Prevention Guidelines.
However, some issues need to be clarified:
1. In the article the authors state that patients received fully prepared frozen preservative and cholesterol free, organic whole food plant-based meals (http://veestro.com) for 5 weeks of 3 meals daily. Since multiple types of meals can be chosen on the online site, it is unclear whether patients choose what to eat, or whether a nutritionist has made a diet plan and the meals have been delivered to patients. If this is the case, please explain and add a table to list dietary intake, including total protein, plant protein, carbohydrate, glycaemic index, total fat, polyinsatured/satured fat, fibers, fruit, vegetable, or any other included food such as legumes, nuts or beans.
2. Is there a plant that has been included in meals more than others? What kind of plant was included in plant-based meals, and among them, are there any that have been reported to have anti-inflammatory, anti-hyperglycaemic, anti-hypercholesterolemic, anti-inflammatory or CVDs protecting effects? For example, it will be interesting to know something about the amount of n-3 PUFA.
3. According to the ACC/AHA Guideline on the Primary Prevention of Cardiovascular Disease, physical activity is also important in reducing CVD risk factors. Did the patients get any type of light or moderate exercise, or did they start exercising during the 5-weeks of the dairy-free vegetarian nutrition intervention? If this is the case, please add the type of activity and time of exercise.
4. Several studies have indicated that healthy HDL-C levels (40–60 mg/dL in men, 50–60 mg/dL in women) are associated with a lower risk of CVD incidence, while low HDL-C levels (<35 mg/dL in each sex) have been associated with an increased risk. Moreover, it has been shown that a healthy diet, characterized by high consumption of fruits, vegetables, legumes, fish, nuts and olive oil, could increase the number of HDL-C particles [1,2]. To date, no specific targets for HDL-C or TG levels have been determined in clinical trials, although increased HDL-C levels predict regression of atherosclerosis while low HDL-C levels are associated with high mortality
in patients with coronary artery disease (CAD), even if associated with low LDL levels. In in this study a significant reduction of HDL-C was observed along with low levels of LDL-C. How this data should be interpreted?
5. In the conclusion of this study the 10-year average risk was reduced from 10.8 to 8.7%. Replacing animal proteins with plant proteins has been found to reduce mortality [3]. In this study, many ASCVD risk scores were reduced, but it will be interesting to investigate the all-cause mortality rate, myocardial infarction, stroke and cardiovascular mortality.
6. In table 1 the authors have shown the characteristics of the recruited patients. It was reported that they are affected by hypertension and diabetes mellitus but it is unclear whether they are taking drugs to control high blood pressure or insulin and glucose levels. Please add.
It will also be important to report if patients have a history of inflammatory disease such as autoimmune disease.
Also, have you analysed other physiological parameters such as body fat percentage or waist circumference at baseline and at the end of the nutritional intervention? If it is the case, please add in table 2.
7. Differences in the pathophysiology, clinical presentation, and management of CVDs were observed between men and women. As noted, among the limitations of the study, the study cohort consists mainly of women as only 7 patients were male. According to the Framingham heart study men experience their first CV event ten years earlier than women. However, this difference is reduced with advancing age, as the risk for CVD in women increases after menopause. Premenopausal women are relatively protected compared to men of the same age. However, this gender gap narrows down after menopause, and the risk is related to the age of onset of menopause. It is possible to include premenopausal or menopausal status in women included in the study? Also, if possible, you should include whether these women are on hormone therapy.
8. According to the American Heart Association, the prevalence of hypertension and CVD among Black/African American women (≥20 years) is 46% and 48%, respectively and health intervention ratings indicate that Black/African American women are less have an advantage than their white counterparts [5]. In order to give more information on nutritional intervention in black women, it is possible to include a subgoup of women, excluding the 7 male, to show if there are significant reductions in cardiovascular risk markers and considering the 10-year ASCVD risk?
References
[1] 2019 ESC/EAS Guidelines for the management of dyslipidaemias: lipid modification to reduce cardiovascular risk: The Task Force for the management of dyslipidaemias of the European Society of Cardiology (ESC) and European Atherosclerosis Society (EAS);
[2] François Mach, Colin Baigent, Alberico L Catapano, Konstantinos C Koskinas, Manuela Casula, Lina Badimon, M John Chapman, Guy G De Backer, Victoria Delgado, Brian A Ference. European Heart Journal, Volume 41, Issue 1, 1 January 2020, Pages 111–188, https://doi.org/10.1093/eurheartj/ehz455;
[3] Mingyang Song, Teresa T. Fung, Frank B. Hu, Walter C. Willett, Valter Longo, Andrew T. Chan, MD, MPH, and Edward L. Giovannucci. Animal and plant protein intake and all-cause and cause-specific mortality: results from two prospective US cohort studies. JAMA Intern Med. 2016; 176(10): 1453–1463. doi: 10.1001/jamainternmed.2016.4182;
[4] Alison G. M. Brown, Linda B. Hudson,1 Kenneth Chui, Nesly Metayer, Namibia Lebron-Torres, Rebecca A. Seguin, and Sara C. Folta. Im proving heart health among Black/African American women using civic engagement: a pilot study. BMC Public Health. 2017; 17: 112. doi: 10.1186/s12889-016-3964-2.
Author Response
Kim Williams et al conducted a non-randomized unblinded intervention 5-week non-dairy vegan diet study among a group of urban African American (AA) volunteers, older than >18y from a community church. Of all 53 volunteers, 44 people adhered to the diet. After 5-weeks markers of cardiovascular risk were significantly improved.
Comment:
- In the Methods the included population is not sufficiently described:
-How and from which population were the volunteers recruited?
-What was the time period (year?) of inclusion?
As indicated, patients were recruited from an urban African American (AA) community church, in which the subjects agreed to eat only the delivered food during a 5-week period, Lent, in the spring of 2019 (pre-pandemic). Please note that in the METHODS section, we state:
A total of 53 volunteers agreed to a 5-week prospective dietary intervention performed during Lent in a predominantly AA church on the south side of Chicago (Trinity United Church of Christ, https://www.trinitychicago.org/).
- Why did the authors not perform a randomized study? Since there is no control group, it is unclear whether positive results were due to the diet intervention study or a more general healthy life style such as quitting smoking, more active lifestyle.
Excellent topic for future investigation. However, the purpose of this study was to only test ACC/AHA nutrition recommendation effects on ASCVD risk. This change is essentially an “n-of-1-study”, before and after, which is a valid method, growing in popularity method to test specific phenomenon, such as the SAMSON trial in NEJM (N-of-1 Trial of a Statin, Placebo, or No Treatment to Assess Side Effects N Engl J Med 2020; 383:2182-2184 DOI: 10.1056/NEJMc2031173).
- Table 1 should include the baseline characteristics of all 53 included volunteers.
We elected to report on those fully participating. Of the 9 “drop outs”, there were actually some who refused to eat the food, subjects who were not forthcoming about their baseline exclusion criteria (e.g., vegetarians), and those who received the 5-weeks of food but were not heard from again, with no follow up data, leaving no way use any of their data, other than reporting the 88% real-world completion rate.
- Statistics: Patients with missing values were omitted from the analyses which could have introduced bias. Instead the authors could have performed multiple imputation to take missing values into account, improve power and reduce bias.
As indicated above, there would not be value in inclusion of missing subjects, since they did not qualify for the study (discovered in retrospect), did not eat the food or get a second set of tests. However, the 88% completion serves as useful data for future planning of studies on similar inner-city populations.

Reviewer 2 Report
Kim Williams et al conducted a non-randomized unblinded intervention 5-week non-dairy vegan diet study among a group of urban African American (AA) volunteers, older than >18y from a community church. Of all 53 volunteers, 44 people adhered to the diet. After 5-weeks markers of cardiovascular risk were significantly improved.
Comment:
- In the Methods the included population is not sufficiently described:
-How and from which population were the volunteers recruited?
-What was the time period (year?) of inclusion?
- - Why did the authors not perform a randomized study? Since there is no control group, it is unclear whether positive results were due to the diet intervention study or a more general healthy life style such as quitting smoking, more active lifestyle.
- Table 1 should include the baseline characteristics of all 53 included volunteers.
- Statistics: Patients with missing values were omitted from the analyses which could have introduced bias. Instead the authors could have performed multiple imputation to take missing values into account, improve power and reduce bias.
Author Response
Response to Reviewer 1:
Dear Authors,
the manuscript presented is well redacted and shown interesting data about a nutrition intervention in a cohort of volunteers in an urban African American community church using the 2019 American College of Cardiology/American Heart Association Primary Prevention Guidelines.
Thank you for this comment.
However, some issues need to be clarified:
1. In the article the authors state that patients received fully prepared frozen preservative and cholesterol free, organic whole food plant-based meals (http://veestro.com) for 5 weeks of 3 meals daily. Since multiple types of meals can be chosen on the online site, it is unclear whether patients choose what to eat, or whether a nutritionist has made a diet plan and the meals have been delivered to patients. If this is the case, please explain and add a table to list dietary intake, including total protein, plant protein, carbohydrate, glycaemic index, total fat, polyinsatured/satured fat, fibers, fruit, vegetable, or any other included food such as legumes, nuts or beans.
Thank you for this suggestion. All of the participants received the same 2 shipments of food as shown in the extensive tables which can be added to the manuscript, if the reviewer prefers this. This is attached as an excel file.
Is there a plant that has been included in meals more than others? What kind of plant was included in plant-based meals, and among them, are there any that have been reported to have anti-inflammatory, anti-hyperglycaemic, anti-hypercholesterolemic, anti-inflammatory or CVDs protecting effects? For example, it will be interesting to know something about the amount of n-3 PUFA.
Thank you for this suggestion. This would be a great addition for future study. In this study, the focus was on following ACC/AHA guidelines on nutrition, which was largely about avoidance of foods associated with CV mortality, including trans- fats, saturated fats, sodium, processed meat, refined grains, cholesterol and sweetened beverages.
According to the ACC/AHA Guideline on the Primary Prevention of Cardiovascular Disease, physical activity is also important in reducing CVD risk factors. Did the patients get any type of light or moderate exercise, or did they start exercising during the 5-weeks of the dairy-free vegetarian nutrition intervention? If this is the case, please add the type of activity and time of exercise.
This would also be a great addition for future study. In this study, the focus was on following ACC/AHA guidelines on nutrition and there was no opportunity to direct, suggest or monitor exercise.
Several studies have indicated that healthy HDL-C levels (40–60 mg/dL in men, 50–60 mg/dL in women) are associated with a lower risk of CVD incidence, while low HDL-C levels (<35 mg/dL in each sex) have been associated with an increased risk. Moreover, it has been shown that a healthy diet, characterized by high consumption of fruits, vegetables, legumes, fish, nuts and olive oil, could increase the number of HDL-C particles [1,2]. To date, no specific targets for HDL-C or TG levels have been determined in clinical trials, although increased HDL-C levels predict regression of atherosclerosis while low HDL-C levels are associated with high mortality
in patients with coronary artery disease (CAD), even if associated with low LDL levels. In in this study a significant reduction of HDL-C was observed along with low levels of LDL-C. How this data should be interpreted?
Consistent with multiple prior publications (e.g., Yoko Yokoyama et al, Association between plant-based diets and plasma lipids: a systematic review and meta-analysis, Nutrition Reviews (2017). DOI: 10.1093/nutrit/nux030), HDL was reduced by reduction of animal product consumption. Yokohama found that:
- In observational studies, a plant-based vegetarian diet is associated with a 22.9 mg/dL reduction in LDL cholesterol and a 3.6 mg/dL reduction in HDL cholesterol, compared to control groups following an omnivorous diet.
- In clinical trials, a plant-based vegetarian diet lowers LDL cholesterol by 12.2 mg/dL and reduces HDL cholesterol by 3.4 mg/dL, compared to control groups following an omnivorous, low-fat, calorie-restricted, or a conventional diabetes diet.
Although in our study HDL was reduced by 11% (from 63 to 56), non-HDL was reduced by 12% (63 to 56 and 127 to 112 mg/dL, respectively, both p=0.000), and consequently ASCVD risk was reduced.
In the conclusion of this study the 10-year average risk was reduced from 10.8 to 8.7%. Replacing animal proteins with plant proteins has been found to reduce mortality [3]. In this study, many ASCVD risk scores were reduced, but it will be interesting to investigate the all-cause mortality rate, myocardial infarction, stroke and cardiovascular mortality.
This would also be a great addition for future study, fully agree. However, the event rate in short-term studies is very small. In this case, there were no events reported in the 5 weeks of the study.
In table 1 the authors have shown the characteristics of the recruited patients. It was reported that they are affected by hypertension and diabetes mellitus but it is unclear whether they are taking drugs to control high blood pressure or insulin and glucose levels. Please add.
Thank you. Of the 44 subjects 25 reported antihypertensive therapy, 11 were on lipid lowering drugs and 7 were on hypoglycemic medications. This has been added to the manuscript and the table.
It will also be important to report if patients have a history of inflammatory disease such as autoimmune disease.
Excellent topic for future studies. These were not obtained in the study of cardiovascular risk factors.
Also, have you analysed other physiological parameters such as body fat percentage or waist circumference at baseline and at the end of the nutritional intervention? If it is the case, please add in table 2.
These were not obtained. Of note, we have extensive clinical experience with waist circumference. It can very emotional for some patients, and be best done in a setting of a physician-patient relationship.
Differences in the pathophysiology, clinical presentation, and management of CVDs were observed between men and women. As noted, among the limitations of the study, the study cohort consists mainly of women as only 7 patients were male. According to the Framingham heart study men experience their first CV event ten years earlier than women. However, this difference is reduced with advancing age, as the risk for CVD in women increases after menopause. Premenopausal women are relatively protected compared to men of the same age. However, this gender gap narrows down after menopause, and the risk is related to the age of onset of menopause. It is possible to include premenopausal or menopausal status in women included in the study? Also, if possible, you should include whether these women are on hormone therapy.
Excellent topic for future investigation. This distinction is not made in the ACC/AHA risk calculator, and this data was not collected.
According to the American Heart Association, the prevalence of hypertension and CVD among Black/African American women (≥20 years) is 46% and 48%, respectively and health intervention ratings indicate that Black/African American women are less have an advantage than their white counterparts [5]. In order to give more information on nutritional intervention in black women, it is possible to include a subgoup of women, excluding the 7 male, to show if there are significant reductions in cardiovascular risk markers and considering the 10-year ASCVD risk?
No, we did not find a difference based on gender.

Round 2
Reviewer 1 Report
Please find the attachment

Author Response
The authors’ answer to question 1 can be considered acceptable.
Thank you.
The authors reported a table relating to the food administered to the study participants, as required. This table must be added as supplementary data.
Thank you. This has been added as a supplementary table at the end, but cannot be formatted with portrait rather than landscape. Thus, the original excel worksheet is included as a separate file for copy editing.
The answer to question 2,3 can be considered acceptable
Thank you.
The answer to question 4 should be included in the discussion of the study.
Thank you. The following text has been added to the discussion:
HDL Cholesterol. Consistent with multiple prior publications, HDL was reduced by reduction of animal product consumption. The meta-analysis by Yokohama et al. [30} found that a plant-based vegetarian diet is associated with a 22.9 mg/dL reduction in LDL cholesterol and a 3.6 mg/dL reduction in HDL cholesterol, compared to control groups following an omnivorous diet. However, in the cited clinical trials, a plant-based diet lowered LDL cholesterol by 12.2 mg/dL and reduced HDL cholesterol by 3.4 mg/dL, compared to control groups following an omnivorous, low-fat, calorie-restricted, or a conventional diabetes diet. Although in our study HDL was reduced by 11% (from 63 to 56), non-HDL was reduced by 12% (63 to 56 and 127 to 112 mg/dL, respectively, both p=0.000), and consequently ASCVD risk was reduced.
The answer to request 5 can be considered acceptable.
Thank you.
The answers to request 6 can be considered acceptable.
Thank you.
The answer to request 7 can be considered acceptable.
Thank you.
Regarding item 8, if the authors found no difference for the 10-year ASCVD risk, it should be shown to the referee.
As noted in the manuscript, the ASCVD risk could not be calculated for 3 subjects with low lipid fractions at baseline and 8 subjects after intervention (p = 0.184), i.e., a total cholesterol < 130 which is not in the acceptable by the risk calculator. Out of the 8 subjects, 3 were men, leaving only 4 men (and 32 women) for the follow up comparison group.
Thus, there was “no significant difference” between males and females since a) the remaining male sample size is small, and b) the most improved males had to be removed because of their excellent response to diet and low total cholesterol levels below 130 mg/dl:
Le 7 sept. 2021 à 14:58, Setri Fugar <[email protected]> a écrit :
Yes no difference- Using a paired T-test.
Males not significant P-value 0.477
Females significant P-value 0.004
Males
Paired Samples Statistics |
|||||
|
Mean |
N |
Std. Deviation |
Std. Error Mean |
|
Pair 1 |
BS ASCVD |
14.45000 |
4 |
10.822969 |
5.411485 |
Followup ASCVD |
14.92500 |
4 |
11.687992 |
5.843996 |
Paired Samples Test |
|||||||||
|
Paired Differences |
t |
df |
Sig. (2-tailed) |
|||||
Mean |
Std. Deviation |
Std. Error Mean |
95% Confidence Interval of the Difference |
||||||
Lower |
Upper |
||||||||
Pair 1 |
BS ASCVD - Followup ASCVD |
-.475000 |
1.172959 |
.586480 |
-2.341440 |
1.391440 |
-.810 |
3 |
.477 |
Females
Paired Samples Statistics |
|||||
|
Mean |
N |
Std. Deviation |
Std. Error Mean |
|
Pair 1 |
BS ASCVD |
10.00312 |
32 |
9.229213 |
1.631510 |
Followup ASCVD |
7.96563 |
32 |
6.593966 |
1.165660 |
Paired Samples Test |
|||||||||
|
Paired Differences |
t |
df |
Sig. (2-tailed) |
|||||
Mean |
Std. Deviation |
Std. Error Mean |
95% Confidence Interval of the Difference |
||||||
Lower |
Upper |
||||||||
Pair 1 |
BS ASCVD - Followup ASCVD |
2.037500 |
3.701939 |
.654417 |
.702809 |
3.372191 |
3.113 |
31 |
.004 |

This manuscript is a resubmission of an earlier submission. The following is a list of the peer review reports and author responses from that submission.
Round 1
Reviewer 1 Report
This manuscripts describes a clinical trial. However, no NCT number or other registration information is available, nor is a CONSORT diagram. The authors describe this as a "cohort;" however, an intervention was provided. The absence of randomization or a control arm does not preclude the need for clinical trial registration. The study prospectively assigned participants to an intervention; it is a clinical trial. I do not agree that this is a cohort study.
Reviewer 2 Report
In the manuscript “Nutrition Intervention for Reduction of Cardiovascular Risk in African Americans Using the 2019 American College of Cardiology/American Heart Association Primary Prevention Guidelines” the Authors tried to determine if implementing the dietary pattern (reduction of dietary sodium, cholesterol, refined carbohydrates, saturated fat and sweetened beverages) could reduce cardiovascular risk. The Authors implemented a 5-week ACC/AHA-styled nutrition intervention, assessed by measuring risk markers and adherence, called HEART-LENS. This is a intervention study, which included a cohort of 53 volunteers. In addition, the Authors assessed only vegetarian meals without dairy products. The study is an important contribution to science due to its subject. The Authors have taken a difficult subject, because this problem affects an increasing number of people worldwide. The study deserves recognition.
Generally the manuscript provides valuable information. However, I have some remarks.
Introduction;
The Authors only described situation of cardiovascular disease in the United States. What is the problem, for example in Europe? A comparison between countries could help to illustrate this problem.
Methods;
The Authors wrote that got approval by Rush University Medical Center Institutional Review Board, but did not write number of approval and date, which got it. The Authors should add this information.
The Authors wrote that use self-reported questionnaire to assess dietary patterns and lifestyle variables. I do not know anything about this questionnaire. What was the questionnaire? What questions included this tool? Was it validated questionnaire? The Authors should explain it.
Why the Authors used such a very low caloric diet? Do not the Authors think that is too small? The calories (1155 kc) are below basal metabolic rate. Please, explain that.
Conclusions;
The Authors repeated results in Conclusions section. In addition, the Authors wrote about intestinal microbiome. It was not subject of the Author’s research. The conclusions should be rewrite.
Line 137-140;
These are not results but description of population. It should be in Methods section
Line 39-40;
Sometimes, the Authors wrote values with one decimal place and sometimes with two, this has to be unified.